# Sequential interleukin-17 inhibitors for moderate-to-severe plaque psoriasis who have an IL-17 inhibitors failure in a resource limited country: An economic evaluation

Piyameth Dilokthornsakul[1,2]*, Ratree Sawangjit[3,4], Nopadon Noppakun[5], Natta Rajatanavin[6], Bensachee Pattamadilok[7], Leena Chularojanamontri[8], Unchalee Permsuwan[1]

1 Department of Pharmaceutical Care, Center for Medical and Health Technology Assessment (CM-HTA), Faculty of Pharmacy, Chiang Mai University, Chiang Mai, Thailand, 2 Department of Pharmacy Practice, Center of Pharmaceutical Outcomes Research, Faculty of Pharmaceutical Sciences, Naresuan University, Mueang, Phitsanulok, Thailand, 3 Clinical Trial and Evidence-Based Synthesis Research Unit (CTEBs RU), Mahasarakham University, Mahasarakham, Thailand, 4 Department of Clinical Pharmacy, Faculty of Pharmacy, Mahasarakham University, Mahasarakham, Thailand, 5 Division of Dermatology, Department of Medicine, Faculty of Medicine, Chulalongkorn University, Pathumwan, Bangkok, Thailand, 6 Division of Dermatology, Faculty of Medicine Ramathibodi Hospital, Mahidol University, Ratchathewi, Bangkok, Thailand, 7 Department of Medical Services, Institute of Dermatology, Ministry of Public Health, Ratchathewi, Bangkok, Thailand, 8 Department of Dermatology, Faculty of Medicine, Siriraj Hospital, Mahidol University, Bangkok Noi, Bangkok, Thailand

* piyameth.dilok@cmu.ac.th, piyamethd@gmail.com

## Abstract

### Background

Biologics has been known to be effective for patients with psoriasis. However, optimal treatment pathways and their cost-effectiveness are limited in a resource-limited country. This study assessed the cost-effectiveness of different sequential biologics for moderate-to-severe plaque psoriasis.

### Method

A hybrid model from a societal perspective was used. Model inputs were derived from network meta-analysis, clinical trials, and published literature. Three different sequential biologic treatments were assessed; Sequence 1; 1st Interleukin-17 (IL-17) inhibitor (secukinumab) followed by 2nd IL-17 inhibitors (ixekizumab or brodalumab), then 3rd IL-23 inhibitor (guselkumab), Sequence 2; ixekizumab followed by secukinumab or brodalumab, then guselkumab, and Sequence 3; brodalumab followed by ixekizumab or secukinumab, then guselkumab. Methotrexate or ciclosporin was used as standard of care (SoC).

### Results

All three different sequential biologic therapies could gain total quality-adjusted life year (QALY), but they had higher cost than SoC. Sequence 1 had the lowest incremental cost-effectiveness ratio (ICER) compared to SoC at 621,373 THB/QALY (19,449 $/QALY). ICER

**Data Availability Statement:** Data used in this study is available at https://doi.org/10.7910/DVN/HN2K7P.

**Funding:** This work was supported by Novartis (Thailand) Limited. The funder had no role in the design and conduct of the study; collection, management, analysis, and interpretation of the data; preparation of the manuscript. The funder reviewed and approved the submission of the manuscript without any major corrections of the final draft of the manuscript.

**Competing interests:** Piyameth Dilokthornsakul receives research grants from Novartis (Thailand) Limited, Pfizer (Thailand) Limited. He also receives honorariums from GSK (Thailand) Limited, and Boehringer Ingelheim (Thailand) Limited. Other authors declare no conflict of interest.

for Sequence 2 was 957,258 THB/QALY (29,962 $/QALY), while that for Sequence 3 was 1,332,262 THB/QALY (41,700 $/QALY). Fully incremental analysis indicated that Sequence 3 was dominated by Sequence 1 and Sequence 2. ICER for Sequence 2 was 7,206,104 THB/QALY (225,551 $/QALY) when compared to Sequence 1.

## Conclusion

At the current willingness-to-pay of 160,000 THB/QALY, no sequential IL-17 inhibitor was cost-effective compared to SoC. Secukinumab followed by ixekizumab or brodalumab then guselkumab (Sequence 1) may be the most appropriate option compared with other treatments.

## 1. Introduction

Psoriasis is a chronic, incurable, skin inflammatory disease. Chronic course and recalcitrant to treatment cause patients to feel distressed and psychological traumatized [1]. Psoriasis impacts patients' work, social lives, and quality of life resulting in a physical and psychological burden [2–4]. Clinical heterogeneity of psoriasis has been classified, but plaque psoriasis is the most common form, accounting for 90% of cases [1]. Topical treatments and targeted phototherapy are commonly used for mild psoriasis, while systemic therapies and total body phototherapy are used for moderate-to-severe psoriasis [5].

Four groups of biologics have been used for psoriasis including tumor necrosis factor-alpha inhibitors, interleukin (IL)-12/IL-23 inhibitor, IL-17 inhibitors, and IL-23 inhibitors. A network meta-analysis indicated that all treatments are more effective than placebo in reaching a 75% or greater reduction in Psoriasis Area and Severity Index scores from baseline (PASI 75). Biologics are more effective than non-biologics and small molecules, especially infliximab, ixekizumab, secukinumab, brodalumab, risankizumab, and guselkumab [6]

Although evidence confirms the efficacy of biologics on psoriasis, the use of biologics is still limited because of its cost [6, 7]. A cost-effectiveness study shows that all biologics could improve quality-adjusted life year (QALY) but they had higher total costs [8]. Even though several cost-effectiveness studies have been published, their findings are not applicable for resource-limited setting because of differences in healthcare system, medication price, and costs of psoriasis treatment [8–11].

In Thailand, three biologics; etanercept, infliximab, and secukinumab, are available for psoriasis patients under the Dermatology Disease Prior Authorization (DDPA) program. It allows patients who failed to conventional systemic therapies under the civil servant medical benefit scheme to receive biologics at no cost. Secukinumab is the only IL-17 inhibitor listed in the DDPA program. Regarding there are differences in patient profile, underlying diseases, and medication history, an individualized treatment, other biologics should be available. This study assessed cost-effectiveness of sequential IL-17 inhibitors for moderate-to-severe plaque psoriasis patients who have an inadequate response to conventional systemic therapies in Thailand.

## 2. Material and methods

### 2.1 Overall description

A cost-utility analysis was conducted from a societal perspective with a lifetime horizon. A decision tree with a Markov model was developed (Fig 1). Patients with moderate-to-severe

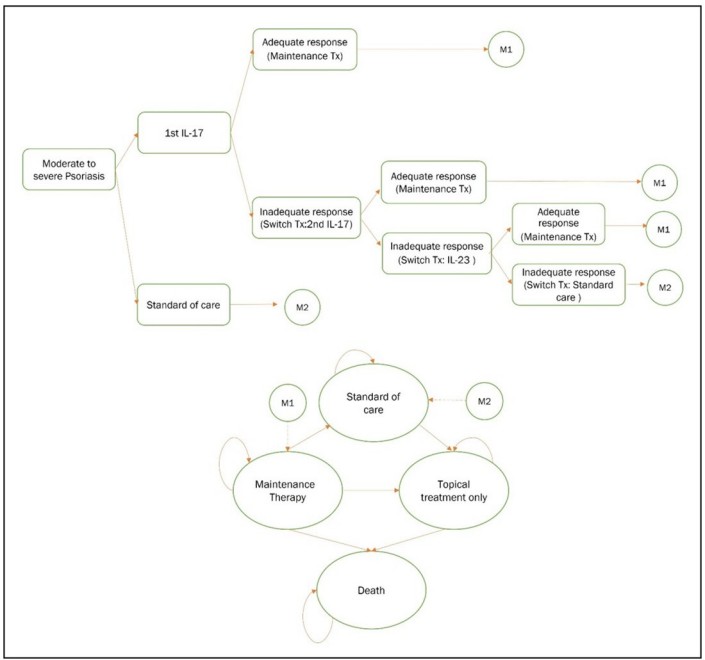

**Fig 1. Schematic diagrams of a hybrid model consisted of a decision tree model along with a Markov model.**

psoriasis aged 40 years old with PASI score of $\geq$ 15 who had an inadequate response to conventional systematic treatments were eligible. According to the Thai HTA guideline [12, 13], two expert meetings were held to determine a scope of study, appropriateness of inputs, analyses, and findings. Four dermatologists and three health economics were involved in the meetings. This study was model-based economic evaluation using all data from existing literature without any additional primary data collection or human subject involvement. Thus, ethical committee approval is not required.

## 2.2 Interventions and comparator

Three sequential biologic treatments were compared to SoC. They consisted of secukinumab, ixekizumab, and brodalumab (IL-17 inhibitors) as the 1st or 2nd treatments, and guselkumab (IL-23 inhibitor) as a rescue medication.

**Sequence 1;** secukinumab 300 mg followed by 300 mg for week 1–4, then 300 mg every 4 weeks for six months, and continue secukinumab 300 mg every 4 weeks if responded. Patients who failed to respond to secukinumab at 6 months received either ixekizumab 80 mg every 4 weeks or brodalumab 210 mg every 2 weeks. Guselkumab 100 mg every 8 weeks was given to patients who failed to respond to the 2nd IL-17 inhibitor.

**Sequence 2;** ixekizumab 160 mg followed by 80 mg every 2 weeks until week 12, then 80 mg every 4 weeks for six months, and continue ixekizumab 80 mg every 4 weeks if responded. Patients who failed to respond to ixekizumab received either secukinumab 300 mg every 4 weeks or brodalumab 210 mg every 2 weeks. Guselkumab 100 mg every 8 weeks was given to patients who failed to respond to the 2nd IL-17 inhibitor.

**Sequence 3;** brodalumab 210 mg was given to the patients at start followed by 210 mg for week 1–2, then 210 mg every 2 weeks for six months, and continue brodalumab 210 mg every 2 weeks if responded. Patients who failed to respond to brodalumab received either

ixekizumab 80 mg every 4 weeks or secukinumab 300 mg every 4 weeks. Guselkumab 100 mg every 8 weeks was given to patients who failed to respond to the 2nd IL-17 inhibitor.

SoC was defined as methotrexate (MTX) 25 mg/week or ciclosporin 5 mg/kg/day. The average weight of 60 kgs was assumed. The proportion of the use of MTX and ciclosporin was 85:15 based on a previous Thai study [14]. Topical treatments (liquor carbonis detergents, tar shampoo, calcipotriol, and topical corticosteroid cream) were applied for all treatments. Patients with no treatment health state were assumed to continue topical treatments.

## 2.3 Model structure and assumptions

Fig 1 shows the model used in this study. In brief, eligible patients received either the 1st IL-17 inhibitor or SoC. Patients who responded to the 1st IL-17 inhibitor at 6 months of treatment moved to maintenance therapy in Markov model, while patients who failed to the 1st IL-17 inhibitor switched to the 2nd IL-17 inhibitor and guselkumab (as a rescue medication, if failed the 2nd IL-17 inhibitor), sequentially. Patients who responded to 2nd IL-17 inhibitor or guselkumab moved to maintenance therapy and continued their biologic treatments, while patients who failed to biologic treatments moved to Markov model at SoC health state.

Some patients with maintenance therapy could drop out and move to SoC. Because of no clear recommendation of the duration of biologic treatments in psoriasis, three years of treatment continuity was assumed. All patients who were in maintenance therapy at the end of year 3 with no drop-out stopped their treatments and moved to topical treatment only health state. After stopping their treatment, patients could have flares and relapse at different relapse rates based on their last biologic and moved to maintenance therapy health state. All patients who had relapsed were assumed to be re-treated with their last biologics and continued for lifelong.

All patients receiving anti IL-17s were assumed to have one-time screening for complete blood count (CBC), fasting blood sugar (FBS), lipid profile, blood urea nitrogen (BUN), and serum creatinine (Cr), and liver function test. Hepatitis B infection and tuberculosis screening were also performed consisting of interferon-gamma release assay (IGRA), chest x-ray (CXR), hepatitis B surface antigen (HBsAg), hepatitis B surface antibody (anti- HBs), and hepatitis B core antibody (anti- HBc). Addition to one-time screening examination, a laboratory monitoring was applied for patients in maintenance therapy. They included bi-yearly CBC, AST, and ALT tests with a yearly IGRA.

Several laboratory examinations were also performed for patients with SoC. They included CBC, AST, ALT, BUN, and Cr every 3–6 months. Lipid profile was also monitored every 3 months. In addition, uric acid and magnesium were monitored quarterly for patients receiving ciclosporin.

Adverse events for biologics were also taken to be accounted. They included serious infection (sepsis and pneumonia) and malignancy. Serious infection was applied at start, while malignancy was applied after 1 year of biologic initiation. Cirrhosis was considered as an adverse event for patients receiving MTX which was the most common medication used as the SoC.

## 2.4 Model validation

The model and its assumptions were clinically validated by dermatologists during the 1st expert meeting. The model's programming was verified by a health economist to ensure the validity of the programming in the model.

## 2.5 Model inputs

**2.5.1 Efficacy, transition probabilities, mortality, and health utility.** A pragmatic literature review was conducted to determine the most updated evidence of treatment response. PubMed database was searched until November 2021. The most updated network meta-analysis reporting the proportion of patients reaching PASI 75 for all included biologics was selected. According to our search, a most updated network meta-analysis [7] which provided sufficient information was selected.

Drop-out and relapse rates were derived from landmark randomized controlled clinical trials [8, 15–24], while rates for serious infection and malignancy were derived from a network meta-analysis [25]. Age-specific mortality for Thais and relative risk of death in patients with psoriasis were used to estimate disease-specific mortality rate [26, 27].

Utilities were derived from previous studies [8, 11, 28]. In addition, baseline PASI score in Thai patients from three studies [29–31] were meta-analyzed and used to estimate utility in patients reaching PASI 75 and PASI 90 (a 90% reduction in PASI score) using a published algorithm by Matza et al. [32] The estimated health utility from baseline PASI in Thai patients was used for a scenario analysis. All inputs are presented in S1 Table.

**2.5.2 Cost.** Direct medical and direct non-medical costs were estimated from a societal perspective. Indirect cost was not included based on the recommendation from Thai health technology assessment guidelines [12, 13] to avoid double-counting. Prices of each biologic, SoC, and topical treatments were provided by pharmaceutical companies, or the reference price published [33]. Resource utilization of different SoC and topical treatments was retrieved from a previous Thai study [14]. Unit costs for laboratory tests were based on the Thai standard cost list [34] and health resource utilization was based on a Thai practice guideline [35].

Direct non-medical cost included additional food cost and transportation cost. Cost/visit was derived from the Thai standard cost list [34], while the average number of visits/year was from a previous study [36]. All costs were inflated to the 2021 value using the consumer price index and converted to US dollar ($) using the average exchange rate in the year 2021 which was 0.0313 $/Thai baht (THB). All estimated costs are presented in S1 Table.

## 2.6 Data analysis

**2.6.1 Cost-effectiveness analysis.** The incremental cost-effectiveness ratio (ICER) using mean or point estimates of each input was calculated as base-case analysis to compare all sequential IL-17 inhibitors with SoC using the formula.

$$\text{ICER} = \frac{\text{Total discounted cost of sequential IL-17 inhibitors} - \text{Total discounted cost of SoC}}{\text{Total discounted QALY of sequential IL-17 inhibitors} - \text{Total discounted QALY of SoC}}$$

The fully incremental analysis was also performed to calculate the ICER of each comparison between sequential IL-17 inhibitors. All analyses applied an annual discount rate of 3% for both cost and QALY [12, 13]. The willingness-to-pay (WTP) of 160,000 THB/QALY (5,008 $/QALY) was used as the cost-effectiveness threshold [12, 13].

A series of one-way sensitivity analysis was conducted to explore the effect of uncertainties around inputs on ICER. The upper and lower bound of 95% confidence intervals or ± 20% from mean or point estimates of each input were utilized in the one-way sensitivity analyses and presented the findings as tornado plots.

Probabilistic sensitivity analysis was also conducted to explore the effect of uncertainties around inputs on ICER using Monte Carlo simulation. The Probabilistic sensitivity analysis randomly selected the value of each input simultaneously and calculated ICER of each iteration for 10,000 iterations and showed the findings as cost-effectiveness analysis plane. The 95%

credible interval of ICER from the simulation was also calculated. In addition, the cost-effectiveness acceptability curve by varying WTP was created.

Additionally, scenario analyses were also performed including: (1) the changes of information sources of health utility value from previous cost-effectiveness studies to the calculation from pooled baseline PASI in Thai patients, (2–3) changes of the definition of treatment response from PASI 75 to PASI 90 and PASI 100, (4–7) changes in a ratio between MTX and ciclosporin from 85:15 to 80:20, 70:30, 0:100, and 100:0 for SoC, (8) changes in MTX from tablet form to injection. All scenario analyses were recommended by experts.

**2.6.2 Budget impact analysis.** A budget impact model was developed to estimate the financial consequences of the adoption of sequential IL-17 inhibitors in lieu of SoC over 5-year time horizon under a payer's perspective. The dynamic cohort model was employed to estimate the number of eligible patients for sequential IL-17 inhibitors. The estimated number of populations aged 40 years or older was approximately 31.6 million [37]. Prevalence, [36] incidence, [38] proportion of patients receiving systemic treatments, [14] and average annual death rate were applied. A 5% uptake rate was assumed. The 91% relapse rate was used for year 4–5 because all patients were assumed to stop their biologic treatments at the end of year 3. All inputs used in budget impact analysis are presented in S2 Table.

## 3. Results

### 3.1 Base-case analysis

All three sequential IL-17 inhibitors provided better QALYs but had higher total cost than SoC. The ICER of Sequence 1 was 621,373 THB/QALY (19,449 $/QALY), while the ICER of Sequence 2 and Sequence 3 were 957,258 THB/QALY (29,962 $/QALY) and 1,332,262 THB/QALY (41,700 $/QALY), respectively (Table 1). Cost of the first biologic was the most contributed cost (S1 Fig)

### 3.2 Fully incremental analysis

Sequence 2 resulted in the highest outcome at 16.649 QALY followed by Sequence 1 at 16.488 QALY and Sequence 2 at 16.318 QALY. In contrast, Sequence 3 had the highest total cost (4,552,700 THB) followed by Sequence 2 (3,808,175 THB) and Sequence 1 (2,646,675 THB). Sequence 3 was dominated by Sequence 2 and Sequence 1. The ICER of Sequence 2 compared to Sequence 1 was 7,206,104 THB/QALY (225,551 $/QALY) (Table 2).

### 3.3 Scenario and sensitivity analyses

Scenario analyses were performed to explore uncertainties around clinical outcomes and data source. All scenario analyses' results indicated that all sequential IL-17 inhibitor treatments

**Table 1. Base-case analysis findings.**

| Treatments | Cost (THB) | Life-year | QALY | Incremental cost (THB) | Incremental cost ($) | Incremental QALY | ICER (THB/QALY) | ICER ($/QALY) |
|---|---|---|---|---|---|---|---|---|
| SoC | 783,388 | 20.589 | 13.489 | Reference | Reference | Reference | Reference | Reference |
| Sequence1 | 2,646,675 | 20.589 | 16.484 | 1,863,287 | 58,321 | 2.995 | 621,373 | 19,449 |
| Sequence2 | 3,808,175 | 20.589 | 16.649 | 3,024,787 | 94,676 | 3.160 | 957,258 | 29,962 |
| Sequence3 | 4,552,700 | 20.589 | 16.318 | 3,769,313 | 117,979 | 2.829 | 1,332,262 | 41,700 |

**Abbreviations:** ICER; incremental cost-effectiveness ratio, QALY; quality-adjusted life year, SoC; standard of care, THB; Thai baht

**Note**: Sequence 1: Secukinumab followed by Ixekizumab/Brodalumab then Guselkumab, Sequence 2: Ixekizumab followed by Secukinumab /Brodalumab then Guselkumab, Sequence 3: Brodalumab followed by Ixekizumab/Secukinumab then Guselkumab

**Table 2. Fully incremental analysis findings.**

| Treatments | Cost (THB) | QALY | ICER (THB/QALY) | ICER ($/QALY) |
|------------|-----------|------|-----------------|---------------|
| **SoC** | 783,388 | 13.489 | Reference | Reference |
| **Sequence3** | 4,552,700 | 16.318 | Dominated | Dominated |
| **Sequence1** | 2,646,675 | 16.484 | 621,373 | 19,449 |
| **Sequence2** | 3,808,175 | 16.649 | 7,206,104 | 225,551 |

**Abbreviations:** ICER; incremental cost-effectiveness ratio, QALY; quality-adjusted life year, SoC; standard of care, THB; Thai baht

**Note**: Sequence 1: Secukinumab followed by Ixekizumab/Brodalumab then Guselkumab, Sequence 2: Ixekizumab followed by Secukinumab /Brodalumab then Guselkumab, Sequence 3: Brodalumab followed by Ixekizumab/Secukinumab then Guselkumab

were not cost-effective for patients with moderate-to-severe psoriasis which were consistent with our base-case analysis findings. The ICER for Sequence 1 ranged from 377,682 to 817,287 THB/QALY ($11,821 - $25,581), while the ICER for Sequence 2 and Sequence 3 ranged from 710,581 to 999,430 THB/QALY ($22,241 –$31,282), and 1,079,665 to 1,373,373 THB/QALY ($33,794–$42,987), respectively. All scenario analyses results are presented in Table 3.

One-way sensitivity analysis showed that utility value for patients who reached PASI 75 was the most influential factor for all sequential IL-17 inhibitors. Other influential factors were utility value for patients who reached PASI 90, response rate of IL-17 inhibitors, response rate of control, and drop-out rate. However, varying the value range did not result in the opposite conclusion from not cost-effective to cost-effective options (S2 Fig).

Probabilistic sensitivity analysis indicated that approximately 98.96–98.98% of iterations for IL-17 inhibitors sequential treatments fell in the upper-right quadrant indicating that all sequential treatment could improve QALY but had higher total cost than SoC (S3 Fig).

The cost-effectiveness acceptability curve showed that no sequential IL-17 inhibitors had a chance to be cost-effective at the current WTP of 160,000 THB/QALY. With the WTP of 600,000 THB/QALY Sequence 1 could reach 50% chance of being cost-effective while there was no chance for Sequence 2 and Sequence 3. (Fig 2)

### 3.4 Budget impact analysis

Based on the assumption that the uptake rate was 5% per year, SoC cost Thai healthcare system ranging from 239–415 million THB /year, while Sequence 1, cost the system approximately 392–590 million THB from Year 1 to Year 5. The budget impact of Sequence 1 was 153 million THB in Year 1 and increased to 175 million THB in Year 5. All budget impact findings are presented in S3 Table.

## 4. Discussion

This study indicated that all sequential IL-17 inhibitor treatments could improve QALY by 2.83–3.16 QALYs but they came with higher total lifetime cost. Consequently, no sequential IL-17 inhibitor treatment was cost-effective.

Sequential biologic treatment is a common practice because of poor response, loss of efficacy, or unacceptable side effects for systemic conventional treatment in patients with moderate-to-severe psoriasis [39–41]. However, guidance of the appropriate second-line of biologic therapies for those who fail to systemic conventional treatment, or the 1st line biologic therapy is still limited [35]. The sequential treatment of IL-17 inhibitors followed by an IL-23 inhibitor were selected because the IL-17 inhibitors are more effective and safety than previous IL-12/23 and TNF-α inhibitors [6]. A previous meta-analysis suggested that that ixekizumab,

**Table 3. Scenario analysis results.**

| Treatments | Cost (THB) | QALY | ICER (THB/QALY) | ICER ($/QALY) |
|---|---|---|---|---|
| **Scenario analysis (1): changes of data source of health utility** | | | | |
| SoC | 783,388 | 13.510 | Reference | Reference |
| Sequence1 | 2,646,675 | 16.748 | 575,435 | 18,011 |
| Sequence2 | 3,808,175 | 16.920 | 886,932 | 27,761 |
| Sequence3 | 4,552,700 | 16.564 | 1,234,020 | 38,625 |
| **Scenario analysis (2): changes of treatment response to PASI90** | | | | |
| SoC | 783,388 | 13.275 | Reference | Reference |
| Sequence1 | 2,898,674 | 16.462 | 663,579 | 20,770 |
| Sequence2 | 3,697,130 | 16.614 | 872,496 | 27,309 |
| Sequence3 | 4,222,512 | 16.360 | 1,114,604 | 34,887 |
| **Scenario analysis (3): changes of treatment response to PASI100** | | | | |
| SoC | 783,388 | 13.224 | Reference | Reference |
| Sequence1 | 2,651,337 | 15.509 | 817,287 | 25,581 |
| Sequence2 | 3,057,814 | 15.647 | 938,463 | 29,374 |
| Sequence3 | 3,272,620 | 15.529 | 1,079,665 | 33,794 |
| **Scenario analysis (4): changes of %MTX use to 80:20** | | | | |
| SoC | 864,715 | 13.489 | Reference | Reference |
| Sequence1 | 2,681,796 | 16.488 | 605,964 | 18,967 |
| Sequence2 | 3,840,216 | 16.649 | 941,660 | 29,474 |
| Sequence3 | 4,591,005 = 8 | 16.318 | 1,317,057 | 41,224 |
| **Scenario analysis (5): changes of %MTX use to 70:30** | | | | |
| SoC | 1,015,321 | 13.489 | Reference | Reference |
| Sequence1 | 2,746,834 | 16.488 | 577,428 | 18,074 |
| Sequence2 | 3,899,550 | 16.649 | 912,775 | 28,570 |
| Sequence3 | 4,661,947 | 16.318 | 1,288,899 | 40,343 |
| **Scenario analysis (6): changes of %MTX use to 100%** | | | | |
| SoC | 563,503 | 13.489 | Reference | Reference |
| Sequence1 | 2,551,719 | 16.488 | 663,034 | 20,753 |
| Sequence2 | 3,721,547 | 16.649 | 999,430 | 31,282 |
| Sequence3 | 4,449,128 | 16.318 | 1,373,373 | 42,987 |
| **Scenario analysis (7): changes of %MTX use to 0% (Ciclosporin only)** | | | | |
| SoC | 2,069,562 | 13.489 | Reference | Reference |
| Sequence1 | 3,202,102 | 16.488 | 377,682 | 11,821 |
| Sequence2 | 4,314,889 | 16.649 | 710,581 | 22,241 |
| Sequence3 | 5,158,524 | 16.318 | 1,091,793 | 34,173 |
| **Scenario analysis (8): changes MTX form from tablet to injection** | | | | |
| SoC | 860,878 | 13.489 | Reference | Reference |
| Sequence1 | 2,646,675 | 16.488 | 595,531 | 18,640 |
| Sequence2 | 3,808,175 | 16.649 | 932,735 | 29,195 |
| Sequence3 | 4,552,700 | 16.318 | 1,304,874 | 40,843 |

**Abbreviations:** ICER; incremental cost-effectiveness ratio, QALY; quality-adjusted life year, SoC; standard of care, THB; Thai baht

**Note**: Sequence 1: Secukinumab followed by Ixekizumab/Brodalumab then Guselkumab, Sequence 2: Ixekizumab followed by Secukinumab /Brodalumab then Guselkumab, Sequence 3: Brodalumab followed by Ixekizumab/Secukinumab then Guselkumab

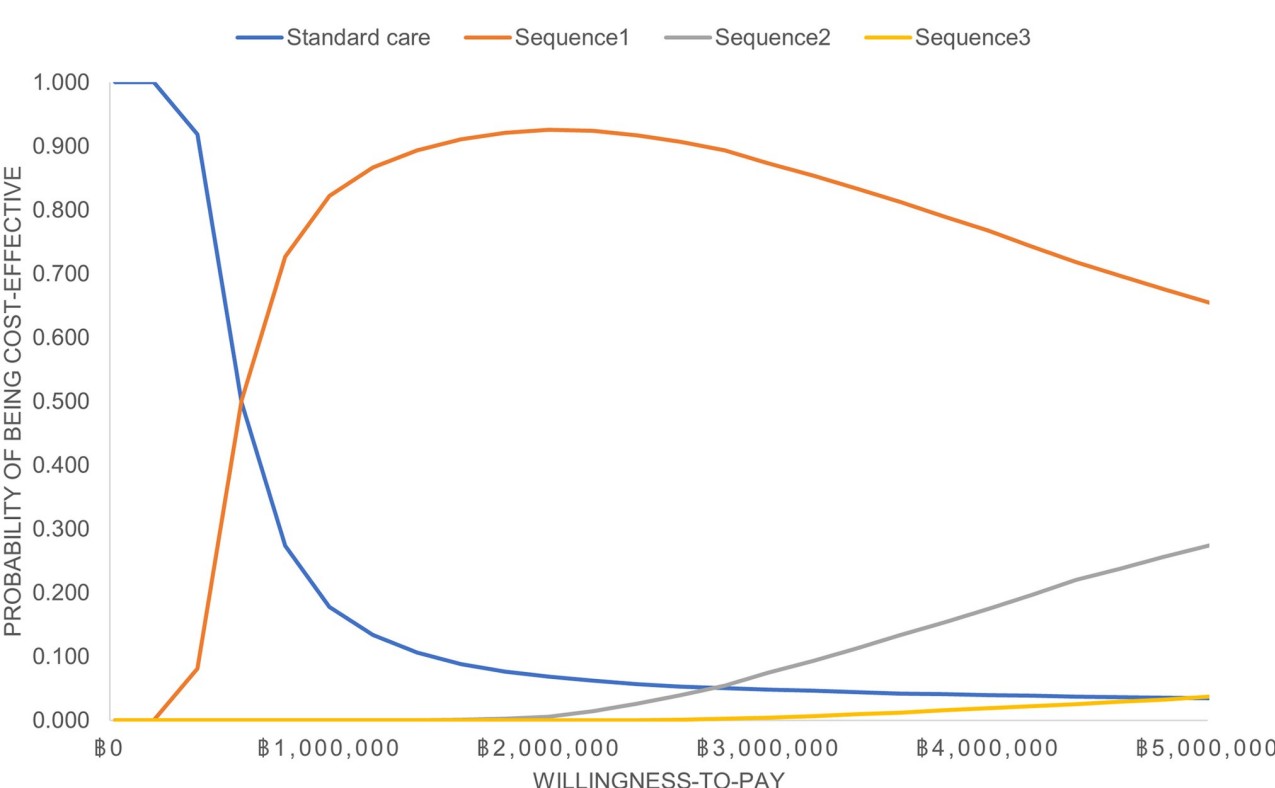

**COST-EFFECTIVENESS ACCEPTABILITY CURVE**

**Fig 2. Cost-effectiveness acceptability curve.** Sequence 1 was secukinumab followed by ixekizumab or brodalumab and guselkumab. Sequence 2 was ixekizumab followed by secukinumab or brodalumab and guselkumab. Sequence 3 was brodalumab followed by secukizumab or ixekizumab and guselkumab.

brodalumab, guselkumab, and secukinumab were associated with the highest PASI response rates [7]. The sequential treatment also reflected the real-world practice in Thailand for patients who were eligible and affordable for biologic treatments.

Although sequential biologic treatments were assumed to provide equal life-years compared SoC, they could improve QALYs (16.318 to 16.649 QALYs) compared to SoC (13.489 QALYs). Our findings were consistent with a previous cost-effective study of IL-17 and TNF-α inhibitors which reported 14.74 to 14.88 QALYs from sequential biologic treatment [42]. These revealed the important benefits of sequential biologic treatment, and it might need to be considered in clinical practice guidelines.

The sequential biologic treatment came with substantial total cost. The incremental cost of each sequence compared to SoC ranged from approximately 1.8–3.7 million THB ($58,400 – $117,0000). The price of the 1st biologic appeared to be the major cost driver. It contributed 69% - 90% of total cost. It was important to decide which biologics should be first initiated. Based on our findings, starting with secukinumab appeared to be a better option because it was the second-best QALY improvement with the lowest total cost. It also had the lowest ICER.

Utility was the most influencing factor for each sequential treatment. We used utility from previous studies [8, 11, 28] which were not conducted in Thai patients. We attempted to derive utility from Thai patients by using a baseline PASI from Thai studies [29–31] and estimated

the utility values using an existing predictive equation provided by Matza et al. [32] A scenario analysis using the utility were consistent with the findings from the base-case analysis. This reflected the robustness of our base-case findings.

We used efficacy data from a previous network meta-analysis [7] which included randomized controlled trials for both biologic-naïve and biologic-experienced patients. Because of the limited evidence in only biologic-experienced patients, we assumed the similar efficacy of biologics in biologic-experienced patients from the analysis [7]. We believed that the assumption was accurate because some previous observational studies [43, 44] showed the similar effects of biologics between biologic-naïve and biologic experienced patients.

Even though psoriasis is a socially and psychologically highly visible disease, clearing the skin flare completely could improve patients' quality of life. A recent network meta-analysis showed that biologic treatment could completely clear or almost clear skin flare (PASI 90) [6]. We still used PASI 75 for assessing the treatment response in our base-case analysis because it has been recommended by several guidelines [35, 45–47] as the treatment response. However, we performed scenario analyses by changing the treatment response definition from PASI 75 to PASI 90 and PASI 100. The findings were also consistent with base-case analysis.

All sequential treatments were not cost-effective at the current WTP. It was because of the price of IL-17 inhibitors. However, they provided better clinical benefits than SoC. The Thai government should consider a price negotiation process to optimize between the price of biologics and their clinical benefits.

## 5. Limitations

Our estimated health resource utilization was based on the disease monitoring recommended by the Thai guideline [35] and expert consensus. It might be different from the actual clinical practice. However, because of the limited evidence of cost of psoriasis in Thailand, we believe this approach was appropriate.

We assumed that biologic treatments did not prolong the length of life. However, psoriasis is not only skin inflammation. It is strongly associated with systemic inflammation such as cardiovascular and psychiatric diseases [5] leading to excess death. Biologics might benefit such diseases and prolong a patient's life. Owing to the limited clinical evidence, we did not include this clinical benefit in our analyses.

## 6. Conclusions

At the current willingness-to-pay, no sequential IL17 inhibitor was cost-effective compared to SoC in Thailand. The possibility of being cost-effective for all sequential treatments was very low compared to SoC. Secukinumab followed by ixekizumab or brodalumab then guselkumab (Sequence 1) seems to be the better option compared with other sequential treatments. Policy makers might need to create strategies to control IL-17 inhibitors price and improve the accessibility to all IL-17 inhibitors as a sequential treatment for patients with moderate-to-severe plaque psoriasis who previously failed one of IL-17 inhibitors in a real-world setting.

### 6.1 Patient and public involvement

Patients or the public were not involved in the design, or conduct, or reporting, or dissemination plans of our research.

## Supporting information

**S1 Table. Cost-utility analysis inputs.**
(PDF)

**S2 Table. Budget impact analysis inputs.**
(PDF)

**S3 Table. Budget impact analysis findings.**
(PDF)

**S1 Fig. Cost contribution (A) % of cost contribution for each sequence, (B) % of cost of each treatment within each seqeunce.**
(PDF)

**S2 Fig. Tornado plots.**
(PDF)

**S3 Fig. Cost-effectiveness analysis plane.**
(PDF)

## Author Contributions

**Conceptualization:** Piyameth Dilokthornsakul, Ratree Sawangjit, Nopadon Noppakun, Bensachee Pattamadilok, Leena Chularojanamontri, Unchalee Permsuwan.

**Data curation:** Piyameth Dilokthornsakul, Ratree Sawangjit, Unchalee Permsuwan.

**Formal analysis:** Piyameth Dilokthornsakul.

**Funding acquisition:** Piyameth Dilokthornsakul.

**Investigation:** Nopadon Noppakun, Natta Rajatanavin, Bensachee Pattamadilok, Leena Chularojanamontri, Unchalee Permsuwan.

**Project administration:** Piyameth Dilokthornsakul.

**Validation:** Ratree Sawangjit, Nopadon Noppakun, Natta Rajatanavin, Bensachee Pattamadilok, Leena Chularojanamontri, Unchalee Permsuwan.

**Writing – original draft:** Piyameth Dilokthornsakul.

**Writing – review & editing:** Piyameth Dilokthornsakul, Ratree Sawangjit, Nopadon Noppakun, Natta Rajatanavin, Bensachee Pattamadilok, Leena Chularojanamontri, Unchalee Permsuwan.

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
