## [Decision Letter · Decision Letter 0]

8 Mar 2024

PONE-D-23-41330Sequential interleukin-17 inhibitors for moderate-to-severe plague psoriasis who have an inadequate response to systemic conventional treatments in a resource limited country: an economic evaluationPLOS ONE

Dear Dr. Dilokthornsakul,

Thank you for submitting your manuscript to PLOS ONE. After careful consideration, we feel that it has merit but does not fully meet PLOS ONE’s publication criteria as it currently stands. Therefore, we invite you to submit a revised version of the manuscript that addresses the points raised during the review process.

Please revise.Please submit your revised manuscript by Apr 22 2024 11:59PM. If you will need more time than this to complete your revisions, please reply to this message or contact the journal office at plosone@plos.org. Please include the following items when submitting your revised manuscript:A rebuttal letter that responds to each point raised by the academic editor and reviewer(s). You should upload this letter as a separate file labeled 'Response to Reviewers'.A marked-up copy of your manuscript that highlights changes made to the original version. You should upload this as a separate file labeled 'Revised Manuscript with Track Changes'.An unmarked version of your revised paper without tracked changes. You should upload this as a separate file labeled 'Manuscript'.

We look forward to receiving your revised manuscript.

Kind regards,

Academic Editor

PLOS ONE

[Piyameth Dilokthornsakul receives research grants from Novartis (Thailand) Limited, Pfizer (Thailand) Limited. He also receives honorariums from GSK (Thailand) Limited, and Boehringer Ingelheim (Thailand) Limited. Other authors declare no conflict of interest.]. 

5. In the online submission form, you indicated that [Data used in this study will be available by appropriate request to the corresponding author. Only data could be provided for non-commercial purposes.]. 

Reviewers' comments:

Reviewer's Responses to Questions

**Comments to the Author**

1. Is the manuscript technically sound, and do the data support the conclusions?

Reviewer #1: Partly

Reviewer #2: Yes

2. Has the statistical analysis been performed appropriately and rigorously? 

Reviewer #1: No

Reviewer #2: Yes

3. Have the authors made all data underlying the findings in their manuscript fully available?

Reviewer #1: Yes

Reviewer #2: Yes

4. Is the manuscript presented in an intelligible fashion and written in standard English?

Reviewer #1: Yes

Reviewer #2: Yes

5. Review Comments to the Author

Reviewer #1: The title of the article need to be improved.

-A main concern is that the manuscript was funded Funding statement:

This work was supported by Novartis (Thailand) Limited. it maybe possible that the Principal Investigator who has the conflicts of interest could be indirectly helping Novartis medication Secukinumab to be approved in Thailand.

The reviewer RESPECTFULLY RECOMMEND TO REQUEST AN OPINION WITH AN STATISTICIAN TO REVIEW THE METHODOLOGY THAT IS SOMEHOW CONFUSING.

Reviewer #2: Title

Correction in title: plaque and not plague

Mhetods

Explain the ethical issues of the study in the methods, including approval by a research ethics committee

Conclusion

I think the conclusions are poor. The results show other things which could be added as conclusions

6. PLOS authors have the option to publish the peer review history of their article (what does this mean?). If published, this will include your full peer review and any attached files.

Reviewer #1: No

Reviewer #2: **Yes: **Marilda AparecidaMilanez Morgado de AbreuDear

---

## [Author Response · Author response to Decision Letter 0]

15 Mar 2024

A rebuttal letter responding to reviewers’ comments point-by-point for a manuscript entitled “Sequential interleukin-17 inhibitors for moderate-to-severe plaque psoriasis who have an inadequate response to systemic conventional treatments in a resource limited country: an economic evaluation”

Comment 1

Response: 

We revised the manuscript format according to the guidelines throughout the manuscript.

Comment 2

Response: 

We have made our model available in the Harvard Dataverse network at this link. https://doi.org/10.7910/DVN/HN2K7P

Comment 3

[Piyameth Dilokthornsakul receives research grants from Novartis (Thailand) Limited, Pfizer (Thailand) Limited. He also receives honorariums from GSK (Thailand) Limited, and Boehringer Ingelheim (Thailand) Limited. Other authors declare no conflict of interest.]. 

Response 3

We confirmed that our conflict of interest does not later our adherence to PLOS ONE policies on sharing data and materials.

 We added it in the Competing interest statement section as shown below.

This does not alter our adherence to PLOS ONE policies on sharing data and materials. Other authors declare no conflict of interest.

Comment 4

Response 4

We have made our model available in the Harvard Dataverse network at this link. https://doi.org/10.7910/DVN/HN2K7P

 We revised the Data sharing statement as shown below.

Data used in this study is available at https://doi.org/10.7910/DVN/HN2K7P

Comment 5 

5. In the online submission form, you indicated that [Data used in this study will be available by appropriate request to the corresponding author. Only data could be provided for non-commercial purposes.]. 

Response 5

We have made our model available in the Harvard Dataverse network at this link. https://doi.org/10.7910/DVN/HN2K7P

 We revised the Data sharing statement as shown below.

Data used in this study is available at https://doi.org/10.7910/DVN/HN2K7P

Comment 6

Response 6

 We revised our manuscript and supplement files according to the guideline.

 

Comments to the Author

Reviewer #1: 

Comment 7

The title of the article need to be improved.

Response 7

We revised our title to make in more concise as shown below.

 Original version

Sequential interleukin-17 inhibitors for moderate-to-severe plaque psoriasis who have an inadequate response to systemic conventional treatments in a resource limited country: an economic evaluation

 Revised version

Sequential interleukin-17 inhibitors for moderate-to-severe plaque psoriasis who previously have an IL-17 inhibitors failure in a resource limited country: an economic evaluation 

Comment 8

-A main concern is that the manuscript was funded Funding statement:

This work was supported by Novartis (Thailand) Limited. it maybe possible that the Principal Investigator who has the conflicts of interest could be indirectly helping Novartis medication Secukinumab to be approved in Thailand.

Response 8

We conducted this study based on the scientific merit not commercial interest. Even though the study was funded by Novartis (Thailand), the funder did not alter our study design, data collection, data analysis, or manuscript writing. The funder only reviewed and approved the submission without any major correction. We originally declared them in the Role of the Funder section. Thus, we don’t think we were attempting to helping secukinumab to be approved in Thailand but providing scientific information on the cost-effectiveness of sequential treatments of anti-IL17 for psoriasis in Thailand for both clinical and policy decisions. 

Comment 9

The reviewer RESPECTFULLY RECOMMEND TO REQUEST AN OPINION WITH AN STATISTICIAN TO REVIEW THE METHODOLOGY THAT IS SOMEHOW CONFUSING.

Response 9

To better explain our statistical analysis, we revised our data analysis section to better explanation as shown below.

 Original version

 2.6.1 Cost-effective analysis 

Base-case analysis was performed, and the incremental cost-effectiveness ratio (ICER) was calculated. All sequential IL-17 inhibitors were compared to SoC. The fully incremental analysis was also performed to compare quality-adjusted life-years (QALYs) and costs among sequential IL-17 inhibitor treatments. An annual discount rate of 3% was applied.[12, 13] 

A series of one-way sensitivity analysis was conducted to explore uncertainties among inputs. The 95% credible interval of each ICER and cost-effectiveness acceptability curve were estimated based on the 10,000 iterations of Monte Carlo simulation. The willingness-to-pay (WTP) threshold of 160,000 THB/QALY (5,008 $/QALY) was used.[12, 13] 

Scenario analyses were also performed including: (1) the changes of information sources of health utility value from previous cost-effectiveness studies to the calculation from pooled baseline PASI in Thai patients, (2 - 3) changes of the definition of treatment response from PASI 75 to PASI 90 and PASI 100, (4-7) changes in a ratio between MTX and ciclosporin from 85:15 to 80:20, 70:30, 0:100, and 100:0 for SoC, (8) changes in MTX from tablet form to injection. All scenario analyses were recommended by experts.

2.6.2 Budget impact analysis

A budget impact model was developed to estimate the financial consequences of the adoption of sequential IL-17 inhibitors in lieu of SoC over 5-year time horizon under a payer’s perspective. The dynamic cohort model was used to estimate the number of eligible patients for sequential IL-17 inhibitors. The estimated number of populations aged 40 years or older was approximately 31.6 million.[37] Prevalence,[36] incidence,[38] proportion of patients receiving systemic treatments,[14] and average annual death rate were applied. A 5% uptake rate was assumed. The 91% relapse rate was used for year 4-5 because all patients were assumed to stop their biologic treatments at the end of year 3. All inputs used in budget impact analysis are presented in supplement at S2 Table. 

Revised version

 2.6.1 Cost-effectiveness analysis 

The incremental cost-effectiveness ratio (ICER) using mean or point estimates of each input was calculated as base-case analysis to compare all sequential IL-17 inhibitors with SoC using the formula.

ICER = Total discounted cost of sequential IL-17 inhibitors - Total discounted cost of SoC

 Total discounted QALY of sequential IL-17 inhibitors - Total discounted QALY of SoC

The fully incremental analysis was also performed to calculate the ICER of each comparison between sequential IL-17 inhibitors. All analyses applied an annual discount rate of 3% for both cost and QALY.[12, 13] The willingness-to-pay (WTP) of 160,000 THB/QALY (5,008 $/QALY) was used as the cost-effectiveness threshold.[12, 13]

A series of one-way sensitivity analysis was conducted to explore the effect of uncertainties around inputs on ICER. The upper and lower bound of 95% confidence intervals or ± 20% from mean or point estimates of each input were utilized in the one-way sensitivity analyses and presented the findings as tornado plots. 

Probabilistic sensitivity analysis was also conducted to explore the effect of uncertainties around inputs on ICER using Monte Carlo simulation. The Probabilistic sensitivity analysis randomly selected the value of each input simultaneously and calculated ICER of each iteration for 10,000 iterations and showed the findings as cost-effectiveness analysis plane. The 95% credible interval of ICER from the simulation was also calculated. In addition, the cost-effectiveness acceptability curve by varying WTP was created. 

Additionally, scenario analyses were also performed including: (1) the changes of information sources of health utility value from previous cost-effectiveness studies to the calculation from pooled baseline PASI in Thai patients, (2 - 3) changes of the definition of treatment response from PASI 75 to PASI 90 and PASI 100, (4-7) changes in a ratio between MTX and ciclosporin from 85:15 to 80:20, 70:30, 0:100, and 100:0 for SoC, (8) changes in MTX from tablet form to injection. All scenario analyses were recommended by experts.

2.6.2 Budget impact analysis

A budget impact model was developed to estimate the financial consequences of the adoption of sequential IL-17 inhibitors in lieu of SoC over 5-year time horizon under a payer perspective. The dynamic cohort model was employed to estimate the number of eligible patients for sequential IL-17 inhibitors. The estimated number of populations aged 40 years or older was approximately 31.6 million.[37] Prevalence,[36] incidence,[38] proportion of patients receiving systemic treatments,[14] and average annual death rate were applied. A 5% uptake rate was assumed. The 91% relapse rate was used for year 4-5 because all patients were assumed to stop their biologic treatments at the end of year 3. All inputs used in budget impact analysis are presented in supplement at S2 Table. 

 

Reviewer #2: Title

Comment 10

Correction in title: plaque and not plague

Response 10

We edited it as suggested.

Comment 11

Methods

Explain the ethical issues of the study in the methods, including approval by a research ethics committee

Response 11

Because this study was model-based economic evaluation. All inputs were from existing literature. No human subjects are involved. Thus, EC approval is not required. We add it in our method as shown below.

Original version 

(none)

Revised version

This study was model-based economic evaluation using all data from existing literature without any additional primary data collection or human subject involvement. Thus, ethical committee approval is not required. 

Comment 11

Conclusion

I think the conclusions are poor. The results show other things which could be added as conclusions.

Response 11

We revised our conclusion to better conclude our findings as shown below.

Original version 

At the current willingness-to-pay, no sequential IL17 inhibitor was cost-effective compared to SoC in Thailand. 

Revised version

At the current willingness-to-pay, no sequential IL17 inhibitor was cost-effective compared to SoC in Thailand. The possibility of being cost-effective for all sequential treatments was very low compared to SoC. Secukinumab followed by ixekizumab or brodalumab then guselkumab (Sequence 1) seems to be the better option compared with other sequential treatments. Policy makers might need to create strategies to control IL-17 inhibitors price and improve the accessibility to all IL-17 inhibitors as a sequential treatment for patients with moderate-to-severe plaque psoriasis who previously failed one of IL-17 inhibitors in a real-world setting.

---

## [Decision Letter · Decision Letter 1]

26 Jun 2024

Sequential interleukin-17 inhibitors for moderate-to-severe plaque psoriasis who have an IL-17 inhibitors failure in a resource limited country: an economic evaluation

PONE-D-23-41330R1

Dear Dr. Dilokthornsakul,

We’re pleased to inform you that your manuscript has been judged scientifically suitable for publication and will be formally accepted for publication once it meets all outstanding technical requirements.

Kind regards,

Academic Editor

PLOS ONE

Additional Editor Comments (optional):

Reviewers' comments:

Reviewer's Responses to Questions

**Comments to the Author**

1. If the authors have adequately addressed your comments raised in a previous round of review and you feel that this manuscript is now acceptable for publication, you may indicate that here to bypass the “Comments to the Author” section, enter your conflict of interest statement in the “Confidential to Editor” section, and submit your "Accept" recommendation.

Reviewer #2: All comments have been addressed

2. Is the manuscript technically sound, and do the data support the conclusions?

Reviewer #2: Yes

3. Has the statistical analysis been performed appropriately and rigorously? 

Reviewer #2: Yes

4. Have the authors made all data underlying the findings in their manuscript fully available?

Reviewer #2: Yes

5. Is the manuscript presented in an intelligible fashion and written in standard English?

Reviewer #2: Yes

6. Review Comments to the Author

Reviewer #2: Dear authors

The suggested changes were made. I think the manuscript is now suitable for publication.

7. PLOS authors have the option to publish the peer review history of their article (what does this mean?). If published, this will include your full peer review and any attached files.

Reviewer #2: **Yes: **Marilda Aparecida Milanez Morgado de Abreu

---

## [Editor Report · Acceptance letter]

1 Jul 2024

PONE-D-23-41330R1 

PLOS ONE

Dear Dr. Dilokthornsakul, 

I'm pleased to inform you that your manuscript has been deemed suitable for publication in PLOS ONE. Congratulations! Your manuscript is now being handed over to our production team.

Kind regards, 

on behalf of

Dr. Robert Jeenchen Chen 

Academic Editor

PLOS ONE